# Event-Triggered Control for Human-in-the-Loop Multi-Agent Systems under DoS Attacks

1st Yuanyuan Xu
*the School of Automation Engineering*
*University of Electronic Science and Technology of China*
Chengdu, China
*the College of Engineering*
*China University of Petroleum-Beijing at Karamay*
Karamay, China
yuanyuanxubhu2016@163.com

2nd Kai Liu
*the College of Engineering*
*China University of Petroleum-Beijing at Karamay*
Karamay, China
15065465226@163.com

3rd Hongjing Liang
*the School of Automation Engineering*
*University of Electronic Science and Technology of China*
Chengdu, China
lianghongjing99@163.com

4th Tieshan Li*
*the School of Automation Engineering*
*University of Electronic Science and Technology of China*
Chengdu, China
*the College of Engineering*
*China University of Petroleum-Beijing at Karamay*
Karamay, China
tieshanli@126.com

5th Yue Long
*the School of Automation Engineering*
*University of Electronic Science and Technology of China*
Chengdu, China
longyue@uestc.edu.cn

6th Qidong Liu
*the School of Automation Engineering*
*University of Electronic Science and Technology of China*
Chengdu, China
liuqidong@std.uestc.edu.cn

7th Ximing Yang
*the School of Automation Engineering*
*University of Electronic Science and Technology of China*
Chengdu, China
yxm961115123@163.com

8th Zongsheng Hang
*the School of Automation Engineering*
*University of Electronic Science and Technology of China*
Chengdu, China
zs_Huang@163.com

*Abstract*—This paper introduces an event-triggered secure control scheme for human-in-the-loop multi-agent systems in the context of DoS attacks. The integration of human intelligence and decision-making significantly enhances system security, as a human provides command signals to a non-autonomous leader agent. To determine unknown states, an adaptive neural state observer utilizes neural networks to approximate nonlinear functions, while a relative threshold-based event-triggered control strategy is introduced to optimize communication resource usage. At the same time, a predictor is developed to monitor potential compromises in the edges of the multi-agent network to counteract attacks. Using Lyapunov analysis, it is shown that the proposed secure control protocol is capable of maintaining bounded closed-loop signals despite the occurrence of attacks. Finally, the effectiveness of the proposed scheme is validated by the simulation results.

*Index Terms*—Multi-agent systems, DoS attacks, event-triggered control, Human in the loop.

This work was supported in part by the National Natural Science Foundation of China under Grant 51939001, Grant 62322307, and Grant 62273072, in part by the Sichuan Science and Technology Program under Grant 2023NSFSC1968, and in part by the 5th Special Funding (Pre-station) of the China Postdoctoral Science Foundation under Grant No. 2023TQ0049. (Corresponding author: Tieshan Li, e-mail: tieshanli@126.com).

## I. INTRODUCTION

IN many earlier studies on coordination control [1]–[5], it is commonly assumed that every agent, including the leader, operates independently within a multi-agent system (MAS). This setup takes advantage of advancements in artificial intelligence, which allows for decreased human involvement. Although it offers a clear benefit, the idea of complete autonomy seems somewhat idealistic when considering the complexities of decision-making in emergencies. This can lead to significant consequences, as evidenced by the documented fatalities associated with Tesla's self-driving cars and the

crashes involving Boeing 737 aircraft [6]. Given the potential for emergencies, it is standard practice to incorporate a human operator to support the autonomous system in performing tasks. Consequently, research on human-in-the-loop (HiTL) control has been carried out, as detailed in [7]–[9].

Generally, network attacks are broadly classified as deception attacks [10] and denial of service (DoS) attacks [11]. In contrast to the injection of malicious information, loss of communication can isolate an agent, thus impeding the effective functioning of multi-agent systems. Taking this into account, experts in control and computational intelligence have made efforts to combat DoS attacks, resulting in successful outcomes. The authors in [12] established sufficient conditions for multi-agent systems under DoS attacks with limited energy. Subsequently, this result was further extended in [13], which also addressed attacks on multiple transmission channels. Similarly, in [14], the authors proposed a recovery mechanism designed to restore communication in network systems during interruptions. Beyond mitigating network attacks, the effective use of communication resources significantly impacts system performance. Event-triggered control is crucial for reducing unnecessary information updates and deserves greater focus. In an event-triggered framework, to avoid unnecessary transmissions, the actuator interacts with the device only when specific trigger conditions are satisfied. In [15], Tabuada *et al.* first introduced an event-triggered control scheme for analyzing nonlinear systems. Following this, a fixed-threshold strategy with a constant upper limit for the trigger condition was implemented in [16] to design a control scheme for linear systems. Although this strategy is simple and easy to implement, it may lead to reduced system performance. To address this issue, In [17], Xing *et al.* advanced the approach from a fixed-threshold to a relative-threshold method, allowing the trigger threshold to be dynamically adjusted according to the magnitude of the trigger error. This approach provides a better balance between system performance and communication resource consumption. Drawing from the previous research, this paper investigates secure control mechanisms within HiTL multi-agent systems using an event-triggered approach, where the control inputs of leaders vary dynamically over time due to human intervention. Unlike existing control schemes [18]–[22], this paper explores a more comprehensive approach, taking into account that inter-agent communication within multi-agent networks may be compromised by malicious attacks. To solve the issue of unavailable state variables, a method for state reconstruction is implemented, alongside an approach that reduces communication frequency and minimizes triggering events by using a threshold-based event-trigger mechanism.

## II. PROBLEM AND FUNDAMENTALS

### A. Graph Theory

A graph is specified by $P = (S, W, A)$, where $S = \{S_1, \ldots, S_{N+M}\}$, with $S_1 = \{1, \ldots N\}$ and $S_{N+M} = \{N + 1, \ldots N + M\}$. The set $W = \{(i, m) \in S \times S\}$ represents the edges, while $A = [a_{im}] \in \mathbb{R}^{(N+M)\times(N+M)}$ denotes the adjacency matrix. These elements are associated with the

sets of nodes, followers, leaders, edges, and the matrix of adjacency in that sequence. Define $F_i$ as the collection of nodes adjacent to node $i$, given by $F_i = \{m|(i, m) \in W\}$. The entries in the adjacency matrix $A$ can be characterized by $a_{im} > 0$ if $(m, i) \in W$; otherwise, $a_{im} = 0$. This implies that $(m, i) \in W$, then agent $m$ obtains data from agent $i$. The scenario where self-connections are absent, indicated by $(i, i) \notin W$, is analyzed. The adjacency matrix element $a_{im}$ is defined as 1 if $(i, i) \in W$, and 0 if $(i, i) \notin W$. The Laplacian matrix $L$ is given by $L = D - A \in \mathbb{R}^{(N+M)\times(N+M)}$, where $D = \text{diag} \{d_1, \ldots d_N\} \in \mathbb{R}^{(N+M)\times(N+M)}$ represents the in-degree matrix, with $d_i = \sum_{m=1, m\neq i}^{N+M} a_{im}$.

### B. System Model

In a MAS with $N$ followers and $M$ leaders, the behavior for each $i$th follower is described by

$$
\begin{aligned}
\dot{\varkappa}_{i,m} &= \varkappa_{i,m+1} + g_{i,m}\left(\bar{\varkappa}_{i,m}\right) \\
\dot{\varkappa}_{i,n} &= u_i + g_{i,n}\left(\bar{\varkappa}_{i,n}\right) \\
y_i &= \varkappa_{i,1}, \\
i &= 1, 2, \ldots, N, m = 1, 2, \ldots, n - 1
\end{aligned}
\tag{1}
$$

where the state vectors are given by $\bar{\varkappa}_{i,m} = [\varkappa_{i,1}, \varkappa_{i,2}, \ldots, \varkappa_{i,m}]^T \in \mathbb{R}^m$, and $y_i \in \mathbb{R}$ denotes the system outputs. The nonlinear functions $g_{i,m}(\cdot)$ and $g_{i,n}(\cdot)$ denote not explicitly known and smooth.

The behavior of the leader is described by

$$
\begin{aligned}
\dot{\varkappa}_{0,j} &= -\varkappa_{0,j} + u_0 \\
\bar{y}_{0,j} &= \varkappa_{0,j}
\end{aligned}
$$

where the states of the leader are represented by $\varkappa_{0,j}$, while $\bar{y}_{0,j}$ denotes the system output, $j = 1, 2$. The control input $u_0$ is an unknown bounded parameter.

The subsequent lemma will be utilized in the design of the adaptive controller.

*Lemma 1:* ( [23]) For any variable $\pi \in \mathbb{R}$, the following inequality holds

$$
0 \leq |\pi| - \frac{\pi^2}{\sqrt{\pi^2 + o^2}} \leq o
$$

where $o > 0$ is a constant.

To identify potential issues, a detector $h(i, m, t)$ has been created to monitor the status of the edge $(i, m) \in W$. When agent $i$ obtains data from agent $m$ at time $t$, $h(i, m, t)$ is set to 0; otherwise, it is set to 1. This mechanism helps agent verify whether the communication link to agent $m$ is compromised.

Therefore, the adjacency matrix $\bar{A} = [\bar{a}_{im}]$, which includes the monitoring information, is expressed as

$$
\bar{a}_{im} = \begin{cases} a_{im}, & h(i, m, t) = 0 \\ 0, & h(i, m, t) = 1 \end{cases}
\tag{2}
$$

## C. Relative Threshold Event-Triggered Control Strategy

To reduce the consumption of communication resources, an adaptive relative threshold-based event-triggered control method [17] was developed.

The following outlines the devised triggering mechanism:

$$u_i(t) = \vartheta_i(t_{i,\chi}), \ \forall t \in [t_{i,\chi}, t_{i,\chi+1}), \chi \in N^* \quad (3)$$

$$t_{i,\chi+1} = \{ \ \inf\{t > t_{i,\chi} \, \|\alpha_i(t)\| \geq \zeta_i |u_i| + \mu_i \} \quad (4)$$

where $\vartheta_i(t)$ represents the controller to be designed subsequently. The decision to update the input $u_i(t)$ is determined by the triggering error $\alpha_i(t) = \vartheta_i(t) - u_i(t)$. The constants $\zeta_i$ and $\mu_i$, which are greater than 0.

From [24], within the time interval $[t_{i,\chi}, t_{i,\chi+1})$, the expression $\vartheta_i(t) = (1 + r_{i,1}(t)\zeta_i) u_i(t) + r_{i,2}(t)\mu_i$ holds, where $|r_{i,1}(t)| \leq 1$ and $|r_{i,2}(t)| \leq 1$. Consequently, the input $u_i(t)$ can be expressed as: $u_i(t) = \frac{\vartheta_i(t)}{1+r_{i,1}(t)\zeta_i} - \frac{r_{i,2}(t)\mu_i}{1+r_{i,1}(t)\zeta_i}$.

Radial basis function neural networks (RBFNNs) are frequently used for modeling nonlinear functions due to their strong ability to approximate functions effectively.

The following assumption is essential for designing the neural state observer.

*Assumption 1:* ( [25]) Assuming function $g(\cdot)$ meets the global Lipschitz condition, specific constants $\varrho_i$ for $i = 1, 2, ..., n$ can be identified to satisfy

$$|g_i(\varkappa_i) - g_i(\hat{\varkappa}_i)| \leq \varrho_i \|\varkappa_i - \hat{\varkappa}_i\| \quad (5)$$

where $\hat{\varkappa}_i = [\hat{\varkappa}_{i,1}, \ldots, \hat{\varkappa}_{i,n}]^T$ represents the estimation of $\varkappa_i$.

## III. EVENT-TRIGGERED SECURE CONTROL SCHEME

Given the system's unobserved states, it is crucial to design an adaptive state estimation approach as outlined below.

$$\dot{\hat{\varkappa}}_{i,m} = \hat{\varkappa}_{i,m+1} + \varrho_{i,m}(y_i - \hat{\varkappa}_{i,1}) + \hat{\beta}_{i,m}^T \omega_{i,m}(\hat{\varkappa}_{i,m})$$

$$\dot{\hat{\varkappa}}_{i,n} = u_i + \varrho_{i,n}(y_i - \hat{\varkappa}_{i,1}) + \hat{\beta}_{i,n}^T \omega_{i,n}(\hat{\varkappa}_{i,n}) \quad (6)$$

where $\hat{\varkappa}_i = [\hat{\varkappa}_{i,1}, \ldots, \hat{\varkappa}_{i,n}]^T$ represents the estimated values of $\varkappa_i$.

Reformulate the the dynamics described in (1)

$$\dot{\varkappa}_i = \bar{\Theta}_i \varkappa_i + \sum_{m=1}^{n} U_{i,m}(g_{i,m}(\bar{\varkappa}_i) + \triangle g_{i,m})$$

$$+ \bar{U}_i u_i + M_i y_i \quad (7)$$

where $\triangle g_{i,m} = g_{i,m}(\bar{\varkappa}_i) - g_{i,m}(\hat{\varkappa}_i)$, $\bar{\Theta}_i = \Theta_i - M_i K_i^T$, and

$$\Theta_i = \begin{bmatrix} 0 & 1 & 0 & \cdots & 0 \\ 0 & 0 & 1 & \cdots & 0 \\ \vdots & \vdots & \vdots & \vdots & \vdots \\ 0 & 0 & 0 & \cdots & 1 \\ 0 & 0 & 0 & \cdots & 0 \end{bmatrix}$$

$$M_i = [\varrho_{i,1}, \varrho_{i,2}, \cdots, \varrho_{i,n}]^T, K_i^T = [1, 0, \cdots, 0]$$

$$U_{i,m} = [0, \cdots, 1, \cdots, 0]^T, \bar{U}_i = [0, \cdots, 0, 1]^T$$

The observer outlined in (6) undergoes reformulation to

$$\dot{\hat{\varkappa}}_i = \bar{\Theta}_i \hat{\varkappa}_i + \sum_{m=1}^{n} U_{i,m}(g_{i,m}(\hat{\varkappa}_i) + \triangle g_{i,m})$$

$$+ \bar{U}_i u_i + M_i y_i \quad (8)$$

Let $e_i = \bar{\varkappa}_i - \hat{\varkappa}_i$ represent the estimation error. The differential of this error is given by

$$\dot{e}_i = \bar{\Theta}_i e_i + \sum_{m=1}^{n} U_{i,m} \left[ \tilde{\beta}_{i,m}^T \omega_{i,m}(\hat{\varkappa}_i) + \varepsilon_{i,m} + \triangle g_{i,m} \right]$$

where $e_i = [e_{i,1}, \ldots, e_{i,n}]^T$. $|\varepsilon_{i,m}| \leq \delta_{i,m}$.

The vector $M_i$ is selected, and the matrix $\bar{\Theta}_i$ is strictly Hurwitz. Consequently, there exists a matrix $\bar{\Xi}_i = \bar{\Xi}_i^T > 0$, and the matrix $\bar{\Psi}_i = \bar{\Psi}_i^T > 0$ satisfies

$$\bar{\Theta}_i^T \bar{\Psi}_i + \bar{\Psi}_i \bar{\Theta}_i = -\bar{\Xi}_i$$

The candidate Lyapunov function $V_e$ is given by

$$V_0 = e_i^T \bar{\Psi}_i e_i$$

The differential of the Lyapunov function $V_0$ can be represented by

$$\dot{V}_0 = -e_i^T \bar{\Xi}_i e_i + 2e_i^T \bar{\Psi}_i \sum_{m=1}^{n} U_{i,m}$$

$$\times \left[ \tilde{\beta}_{i,m}^T \omega_{i,m}(\hat{\varkappa}_i) + \varepsilon_{i,m} + \triangle G_i \right] \quad (9)$$

By evaluating

$$2e_i^T \bar{\Psi}_i \sum_{m=1}^{n} U_{i,m} \left[ \tilde{\beta}_{i,m}^T \omega_{i,m}(\hat{\varkappa}_i) + \varepsilon_{i,m} \right]$$

$$\leq 2(n+1)e_i^T e_i + \|\bar{\Psi}_i\|^2 \sum_{m=1}^{n} \left( \tilde{\beta}_{i,m}^T \tilde{\beta}_{i,m} + \delta_{i,m}^2 \right) \quad (10)$$

$$2e_i^T \bar{\Psi}_i \sum_{m=1}^{n} U_{i,m} \triangle g_{i,m}$$

$$\leq \left( n + \|\bar{\Psi}_i\|^2 \sum_{m=1}^{n} \varrho_{i,m}^2 \right) e_i^T e_i \quad (11)$$

Substituting (11), (12) into (10), it can be determined that

$$\dot{V}_0 \leq -\left( \lambda_{\min}(\bar{\Xi}_i) - 3n - 2 - \|\bar{\Psi}_i\|^2 \sum_{m=1}^{n} \varrho_{i,m}^2 \right) e_i^T e_i$$

$$+ \|\bar{\Psi}_i\|^2 \sum_{m=1}^{n} \left( \tilde{\beta}_{i,m}^T \tilde{\beta}_{i,m} + \delta_{i,m}^2 \right)$$

The coordinate transformations can be specified in the following way:

$$z_{i,1} = \sum_{m=1}^{N} \bar{a}_{im}(\varkappa_{i,1} - \varkappa_{m,1}) + \sum_{m=N+1}^{N+M} \bar{a}_{im}(\varkappa_{i,1} - y_{j,r})$$

$$z_{i,m} = \hat{\varkappa}_{i,m} - \theta_{i,m} \quad (12)$$

where the dynamics errors are denoted by $z_{i,m}$ (for $m = 2, \ldots, n$), while $\theta_{i,m}$ denotes the filter's output.

The outputs $y_{jr}$, $\dot{y}_{j,r}$ of multiple leaders are bounded in their trajectories.

The following expression represents the error compensation signal:

$$
\begin{aligned}
\dot{\rho}_{i,1} &= -c_{i,1}\rho_{i,1} + \varpi_{i,1}\left(\theta_{i,2} - \tau_{i,1} + \rho_{i,2}\right) \\
\dot{\rho}_{i,2} &= -c_{i,2}\rho_{i,2} + \left(\theta_{i,3} - \tau_{i,2}\right) \\
&\quad -\varpi_{i,1}\rho_{i,1} + \rho_{i,3} \\
\dot{\rho}_{i,m} &= -c_{i,m}\rho_{i,m} + \left(\theta_{i,m+1} - \tau_{i,m}\right) \\
&\quad -\rho_{i,m-1} + \rho_{i,m+1} \\
\dot{\rho}_{i,n} &= 0
\end{aligned}
\tag{13}
$$

where $m = 1, \cdots, n$. The constants $c_{i,1}, c_{i,2}, \ldots, c_{i,m} > 0$. $\rho_{i,m}(0)$ set to 0 for $m$ from 1 to $n$. $\varpi_{i,1} = \sum_{m=1}^{N+M} \bar{a}_{im}$. Details regarding the virtual controllers $\tau_{i,1}, \tau_{i,2}, \ldots, \tau_{i,m}$ will be presented in later sections.

The formulation for the error compensation tracking signal is:

$$
\kappa_{i,m} = z_{i,m} - \rho_{i,m}, \quad m = 1, \ldots, n
\tag{14}
$$

Design the virtual controller $\tau_{i,1}$, along with the adaptive laws $\dot{\hat{\beta}}_{i,1}$ and $\dot{\hat{\vartheta}}_{i,1}$

$$
\begin{aligned}
\tau_{i,1} &= -c_{i,1}z_{i,1} - 2\varpi_{i,1}\kappa_{i,1} - \hat{\beta}_{i,1}^T\omega_{i,1} \\
&\quad + \frac{1}{\varpi_{i,1}}\hat{\vartheta}_{i,1}^T\Upsilon_i + \dot{y}_{j,r} - \frac{1}{2\varpi_{i,1}}\kappa_{i,1}
\end{aligned}
\tag{15}
$$

$$
\dot{\hat{\beta}}_{i,1} = l_{i,1}\varpi_{i,1}\omega_{i,1}\kappa_{i,1} - \varsigma_{i,1}\hat{\beta}_{i,1}
\tag{16}
$$

$$
\dot{\hat{\psi}}_{i,1} = -\bar{c}_{i,1}\Upsilon_i\kappa_{i,1} - \bar{\varsigma}_{i,1}\hat{\psi}_{i,1}
\tag{17}
$$

where $\beta_{i,1}$ and $\vartheta_{i,1}$ are the weight vectors. $\omega_{i,1}$ and $\Upsilon_i$ are the basis function vectors. The constants $c_{i,1}$, $l_{i,1}$, $\varsigma_{i,1}$, $\bar{c}_{i,1}$, $\bar{\varsigma}_{i,1} > 0$.

Design the virtual controller $\tau_{i,m}$, along with the adaptive law $\dot{\hat{\beta}}_{i,m}, m = 2, \ldots n-1$.

$$
\begin{aligned}
\tau_{i,m} &= -c_{i,m}z_{i,m} - z_{i,m-1} - \hat{\beta}_{i,m}^T\omega_{i,m} \\
&\quad - \varrho_{i,m}e_{i,1} + \dot{\theta}_{i,m}
\end{aligned}
\tag{18}
$$

$$
\dot{\hat{\beta}}_{i,m} = l_{i,m}\phi_{i,m}\kappa_{i,m} - \varsigma_{i,m}\hat{\beta}_{i,m}
\tag{19}
$$

where $\beta_{i,m}$ is the weight vector. $\omega_{i,m}$ is the basis function vectors. The constants $c_{i,m}$, $l_{i,m}$, $\varrho_{i,m}$, $\varsigma_{i,m} > 0$.

Design the virtual controller $\tau_{i,n}$, along with the adaptive laws $\dot{\hat{\beta}}_{i,n}$ and $\dot{\hat{\varphi}}_i$

$$
\tau_{i,n} = -c_{i,n}z_{i,n} - \frac{1}{2a_i^2}\rho_i\kappa_{i,n}\hat{\varphi}_{i,n}\varsigma_{i,n}^T\varsigma_{i,n}
\tag{20}
$$

$$
\dot{\hat{\beta}}_{i,n} = l_{i,n}\omega_{i,n}\kappa_{i,n} - \varsigma_{i,n}\hat{\beta}_{i,m}
\tag{21}
$$

$$
\dot{\hat{\varphi}}_i = \frac{\sigma_i}{2a_i^2}\kappa_{i,n}^2\varsigma_{i,n}^T\varsigma_{i,n} - \bar{\varsigma}_{i,n}\hat{\varphi}_i
\tag{22}
$$

where the constants $c_{i,n}$, $a_i$, $l_{i,n}$, $\varsigma_{i,n}$, $\rho_i$, $\bar{\varsigma}_{i,n} > 0$.

Design the adaptive controller in the following manner:

$$
\begin{aligned}
\vartheta_i(t) &= -(1+\xi_i)\left(\kappa_{i,n}\tanh\left(\frac{\lambda_{i,n}\kappa_{i,n}}{\epsilon_i}\right)\right. \\
&\quad \left. + \bar{\gamma}_i\tanh\left(\frac{\bar{\gamma}_i\kappa_{i,n}}{\epsilon_i}\right)\right)
\end{aligned}
\tag{23}
$$

with $\bar{\gamma}_i$, $\epsilon_i > 0$.

## IV. STABILITY ANALYSIS

*Theorem 1:* Consider the multi-agent systems (1) subject to DoS attacks, considering Assumptions 1 and 2. The virtual controllers (15), (18), (20), the event-triggered adaptive controller (23), the observer (6), and the adaptive laws (16), (17), (19), (21), and (22) provides assurance that, despite the occurrence of DoS attacks, all closed-loop signals will stay within bounded limits.

**Proof.** Establish the total Lyapunov function as

$$
V = \sum_{i=1}^{N} V_e + \sum_{i=1}^{N} V_{i,n}
$$

The differential of $V$ fulfills

$$
\begin{aligned}
\dot{V} &\leq -\sum_{i=1}^{N}\left(\lambda_{\min}\left(\bar{\Xi}_i\right) - 3n - \frac{5}{2} - \left\|\bar{\Psi}_i\right\|^2\sum_{m=1}^{n}\varrho_{i,m}^2\right) \\
&\quad \times e_i^T e_i + \sum_{i=1}^{N}\sum_{m=1}^{n}\delta_{i,m}^2 - \sum_{i=1}^{N}\sum_{s=2}^{n}\left(c_{i,s}-1\right)\kappa_{i,s}^2 \\
&\quad + \sum_{i=1}^{N}\sum_{m=1}^{n}\frac{\varsigma_{i,s}}{2l_{i,s}}\beta_{i,s}^T\beta_{i,s} + \sum_{i=1}^{N}\frac{\bar{\varsigma}_{i,1}}{2\bar{c}_{i,1}}\psi_{i,1}^T\psi_{i,1} \\
&\quad + \frac{1}{2}\sum_{i=1}^{N}\left(\eta_{i,1}^2 + \iota_{i,1}^2 + a_i^2 + \bar{h}_{i,n}^2\right) \\
&\quad + \sum_{i=1}^{N}\frac{\rho_i}{2\sigma_i}\bar{\varsigma}_{i,n}\varphi_i^2 - \sum_{i=1}^{N}\sum_{s=2}^{n}\left(\frac{1}{4}\right)\tilde{\beta}_{i,s}^T\tilde{\beta}_{i,s} \\
&\quad - \sum_{i=1}^{N}\frac{\rho_i}{2\sigma_i}\bar{\varsigma}_{i,n}\tilde{\varphi}_i^2 - \sum_{i=1}^{N}\frac{\bar{\varsigma}_{i,1}}{2\bar{c}_{i,1}}\tilde{\psi}_{i,1}^T\tilde{\psi}_{i,1} \\
&\quad - \sum_{i=1}^{N}\sum_{s=1}^{n}\left(\frac{\varsigma_{i,s}}{2l_{i,s}} - \left\|\bar{\Psi}_i\right\|^2\right)\tilde{\beta}_{i,s}^T\tilde{\beta}_{i,s} \\
&\quad - \sum_{i=1}^{N}c_{i,1}\kappa_{i,1}^2 + \sum_{i=1i}^{N}2o_i
\end{aligned}
\tag{24}
$$

Utilizing the aforementioned analysis, the results in (24) can be expressed as

$$
\dot{V} \leq HV + Y
\tag{25}
$$

where

$$H = min \left\{ \frac{\lambda_{\min}(\bar{\Xi}_i) - 3n - \frac{5}{2} - \|\bar{\Psi}_i\|^2 \sum_{m=1}^{n} \varrho_{i,m}^2}{\lambda_{\min}(\bar{\Psi}_i)}, \right.$$
$$2c_{i,1}, 2(c_{i,2} - 1), \cdots, 2(c_{i,n} - 1),$$
$$\left. 2l_{i,s} \left( \frac{\varsigma_{i,s}}{2a_{i,s}} - \|\bar{\Psi}_i\|^2 - \frac{1}{4} \right), \bar{\varsigma}_{i,1}, \bar{\varsigma}_{i,n} \right\}$$

$$Y = \sum_{i=1}^{N} \sum_{m=1}^{n} \delta_{i,m}^2 + \sum_{i=1}^{N} \sum_{m=1}^{n} \frac{\varsigma_{i,s}}{2l_{i,s}} \beta_{i,s}^T \beta_{i,s} + \sum_{i=1i}^{N} 2o_i$$
$$+ \frac{1}{2} \sum_{i=1}^{N} \left( \eta_{i,1}^2 + \iota_{i,1}^2 + a_i^2 + \bar{h}_{i,n}^2 \right)$$
$$+ \sum_{i=1}^{N} \frac{\bar{\varsigma}_{i,1}}{2\bar{r}_{i,1}} \psi_{i,1}^T \psi_{i,1} + \sum_{i=1}^{N} \frac{\rho_i}{2\sigma_i} \bar{\varsigma}_{i,n} \varphi_i^2$$

From [26], it follows that $\|\theta_{i,m+1} - \tau_{i,m}\| \leq \bar{\Re}_i$. As a result, $\|\theta_{i,m}\|$ will be constrained, ensuring that all signals stay bounded even when DoS attacks occur. Theorem 1 has been fully proved. ∎

From [24], it can be concluded that

$$\frac{d}{dt} |\alpha_i(t)| = \frac{d}{dt} (\alpha_i(t) \times \alpha_i(t))^{\frac{1}{2}}$$
$$= sign(\alpha_i(t)) \dot{\alpha}_i(t) \leq \left| \dot{\vartheta}(t) \right|$$

where $\left| \dot{\vartheta}(t) \right| \leq \sigma$, $\sigma > 0$.

Based on (3), (4) and $\alpha_i(t)$, it can be determined that $|\alpha_i(t)| \geq \zeta_i |u_i| + \mu_i$. There exists a time interval $t^*$ such that $t_{i,\chi+1} - t_{i,\chi} \geq t^*$. Consequently, the lower bound for the interval between executions $t^*$ satisfies $t^* \geq \frac{\zeta_i |u_i| + \mu_i}{\sigma}$, thus successfully avoiding Zeno behavior.

## V. SIMULATION

In this section, simulation examples will be provided to validate the theoretical results.

Investigate the dynamics related to the $i$th agent in the following manner:

$$\dot{\varkappa}_{i,1} = \varkappa_{i,2} + g_{i,1}$$
$$\dot{\varkappa}_{i,2} = u_i + g_{i,2}$$
$$i = 1, 2, 3$$

where

$$g_{1,1} = \cos(\varkappa_{1,1}), g_{1,2} = \cos(\varkappa_{1,1}\varkappa_{1,2})$$
$$g_{2,1} = \sin(0.5\varkappa_{2,1}), g_{2,2} = \sin(\varkappa_{2,1}\varkappa_{2,2})$$
$$g_{3,1} = \sin(0.2\varkappa_{3,1}), g_{3,2} = \sin(\varkappa_{3,1}\varkappa_{3,2})$$

the state vectors and control input are given by $\varkappa_i = [\varkappa_{i,1}, \varkappa_{i,2}]^T$ and $u_i$, respectively.

The dynamics of the leader can be described by the following equation:

$$\dot{\varkappa}_{0,j} = -\varkappa_{0,j} + u_0$$
$$\bar{y}_{0,j} = \varkappa_{0,j}$$

TABLE I
THE INITIAL STATES OF THE FOLLOWERS AND THEIR CORRESPONDING
STATE ESTIMATIONS.

| Initial states | Values | Initial states | Values |
|---|---|---|---|
| $\varkappa_{1,1}(0)$ | 1.5 | $\hat{\varkappa}_{1,1}(0)$ | 0.5 |
| $\varkappa_{1,2}(0)$ | -0.5 | $\hat{\varkappa}_{1,2}(0)$ | 0.5 |
| $\varkappa_{2,1}(0)$ | -0.7 | $\hat{\varkappa}_{2,1}(0)$ | 0.5 |
| $\varkappa_{2,2}(0)$ | 0.3 | $\hat{\varkappa}_{2,2}(0)$ | 0.5 |
| $\varkappa_{3,1}(0)$ | 1.4 | $\hat{\varkappa}_{3,1}(0)$ | 1.5 |
| $\varkappa_{3,2}(0)$ | -0.3 | $\hat{\varkappa}_{3,2}(0)$ | 0.5 |

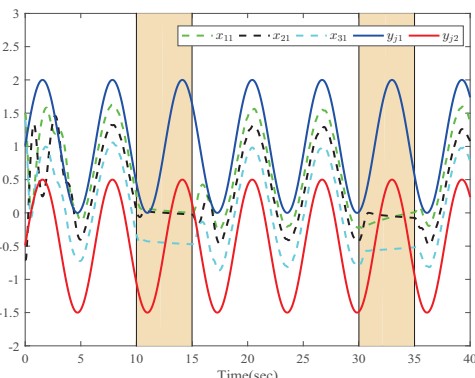

Fig. 1. Tracking of followers, leaders and DoS attacks.

where $\varkappa_{0,j}$ represents the states of the leader, and $\bar{y}_{0,j}$ denotes the system output, $j = 1, 2$. For containment control purposes, the outputs are defined as $y_{j,1} = \bar{y}_{0,1} + 0.5$ and $y_{j,2} = \bar{y}_{0,2} - 0.5$. Additionally, the control input $u_0$ is an unknown bounded variable. The initial conditions are specified in Table I.

Moreover, the control input of the leader, represented as $u_0(t)$ and provided by a human operator, is not available to all followers. The design of the control input $u_0(t)$ is outlined as follows:

$$u_0(t) = \begin{cases} \sin^2(t), 0 \leq t \leq 15 \\ \sin(t)\cos(t), 15 < t \leq 30 \\ \cos^2(t), t > 30 \end{cases}$$

Fig. 1 illustrates the trajectories of followers $\varkappa_{i,1}$ for $i = 1, 2, 3$ and leaders $y_{j,r}$ for $r = 1, 2$ during the DoS attacks. The shaded flesh area indicates the period of the DoS attacks. With the implementation of the proposed secure control scheme, the system returns to normal operation after a brief period. Fig. 2 depicts the timing and intervals of the event-triggered communication strategy.

## VI. CONCLUSION

This paper has presented a secure control scheme for HiTL MASs utilizing an event-triggered mechanism to address DoS attacks. By integrating human intelligence and decision-making, the system's security has been significantly improved as human operators send command signals to a non-autonomous leader agent. To manage limited communication resources, a relative threshold event-triggered control strategy has been adopted, which effectively reduces data transmissions. Simultaneously, the system's vulnerability to malicious

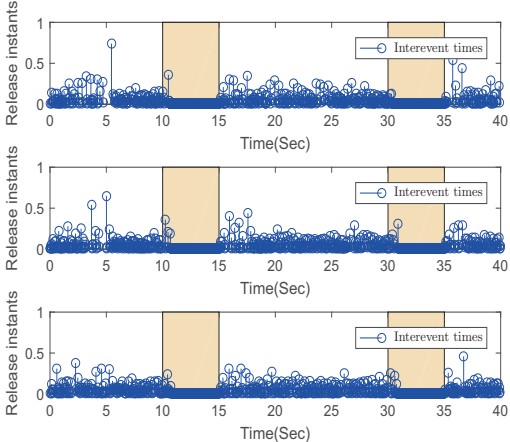

Fig. 2. Intersample time of agents 1-3.

attacks in the network, which could compromise control tasks, is acknowledged. Therefore, the effects of DoS attacks are considered.

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
