# OpenReview forum: "$Event-Triggered Control for Human-in-the-Loop Multi-Agent Systems under DoS Attacks$"
_IEEE.org/ICIST/2024/Conference — IEEE ICIST 2024 Conference Submission_

### Official Review · Reviewer_br8Y · 2024-09-01
**Comments to paper 52**

**Rating:** 10
**Confidence:** 5

**Review:**

This paper introduces an event-triggered secure control scheme for human-in-the-loop multi-agent systems in the context of DoS attacks. The effectiveness of the proposed scheme is validated by the simulation results. The presented results are impressive. The paper can be accepted for publication after some necessary revisions. The following issues are suggested for improving the paper:
1. The controller has many parameters. Please give a remark to illustrate the effect of the controller
parameters.
2. Some future works are also suggested in the conclusion section.
3. Some typos and grammar errors should be modified in this paper. Please carefully double check the manuscript.

---

### Official Review · Reviewer_CY6w · 2024-09-01
**Please the author double-check this paper.**

**Rating:** 6
**Confidence:** 5

**Review:**

1. Please clarify the role of human-in-the-loop control when the system is under attack.
2. There is an incorrect description in Part A of Section II.
3. The authors mentioned human-in-the-loop control in the abstract and Section I, but the designed human-in-the-loop control cannot be found in Section II.
4. The authors have not modeled DoS attacks in this paper and have not specified which term of the proposed controller is designed to counteract them.
5. The authors should explicitly outline the motivation and contributions of this paper in comparison to existing work in the field。
6. Many symbols have not been defined.

---

### Official Review · Reviewer_njBs · 2024-09-02
**This paper can be accepted.**

**Rating:** 7
**Confidence:** 5

**Review:**

This paper investigates an event-triggered secure control scheme for human-in-the-loop multi-agent systems in the context of DoS attacks.
However in the reviewer's opinion, there are somecomments in the paper which should be addressed by the authors:
1. In addition to the contribution of the proposed work, it is suggested to discuss its main limitations/shortcomings.
2. Please add a structural flowchart of the paper's framework to enhance the readability of the paper.
3.Muhc more simulation results are needed to validate the effectiveness of the proposed method in the paper..
4.The English grammar and format of this paper could be further polished and checked carefully.

---

### Decision · Program_Chairs · 2024-09-06

Accept (Oral)